# QTLs Related to Rice Callus Regeneration Ability: Localization and Effect Verification of *qPRR3*

**DOI:** 10.3390/cells11244125

**Published:** 2022-12-19

**Authors:** Jiemin Wu, Xinlei Chang, Chuanhong Li, Zhaoyang Zhang, Jianguo Zhang, Changxi Yin, Weihua Ma, Hao Chen, Fei Zhou, Yongjun Lin

**Affiliations:** 1National Key Laboratory of Crop Genetic Improvement, National Centre of Plant Gene Research, Wuhan 430070, China; 2College of Life Science and Technology, Huazhong Agricultural University, Wuhan 430070, China; 3College of Plant Science and Technology, Huazhong Agricultural University, Wuhan 430070, China

**Keywords:** callus, callus regeneration ability, indica rice, plant regeneration rate, QTL, recombinant inbred line, tissue culture, total green plant rate

## Abstract

Mature and efficient tissue culture systems are already available for most japonica rice varieties (*Oryza sativa* ssp. *geng*). However, it remains challenging to regenerate the majority of indica rice varieties (*Oryza sativa* ssp. *xian*). In this study, quantitative trait loci (QTLs) associated with rice callus regeneration ability were identified based on the plant regeneration rate (PRR) and total green plant rate (TGPR) of the 93-11 × Nip recombinant inbred line population. Significant positive correlations were found between PRR and TGPR. A total of three QTLs (one for PRR and two for TGPR) were identified. *qPRR3* (located on chromosome 3) was detected for both traits, which could explain 13.40% and 17.07% of the phenotypic variations of PRR and TGPR, respectively. Subsequently, the effect of *qPRR3* on callus regeneration ability was validated by cryptographically tagged near-isogenic lines (NILs), and the QTL was narrowed to an interval of approximately 160 kb. The anatomical structure observation of the regenerated callus of the NILs revealed that *qPRR3* can improve the callus regeneration ability by promoting the regeneration of shoots.

## 1. Introduction

Rice is one of the most important crops in the world and a model monocotyledon for studying genetic and functional genomes. Over the past few decades, great efforts have been made to improve the quality and yield of rice. Traditional breeding techniques mainly involve the indirect selection of genotypes through phenotypes, which is more suitable for qualitative traits but relatively difficult for quantitative traits. With the completion of rice genome sequencing and in-depth studies of the functional genome, molecular design breeding and transgenic breeding have developed rapidly, which have more definite targets and higher accuracy, precision, and efficiency.

Transgenic and functional genomics research in rice heavily relies on tissue culture and genetic transformation systems. For japonica rice (*Oryza sativa* ssp. *geng*), N6 basal medium [1] is widely used, and efficient tissue culture and genetic transformation systems are already available [2,3,4,5,6,7]. However, for indica rice (*Oryza sativa* ssp. *xian*), genetic transformation remains a great challenge despite the great efforts that have been made by researchers [8,9,10,11,12,13,14,15,16]. Different rice genotypes face different challenges in tissue culture. Indica varieties have larger genotypic differences between them, and, as a result, their tissue culture and genetic transformation systems are not as widely adaptable as those of japonica varieties [9].

In plants, traits related to tissue culture can be either qualitative [17] or quantitative [18,19]. Quantitative trait loci (QTL) mapping can assess and map the loci controlling the genetic variations responsible for complex traits [20,21]. In the past few decades, various traditional molecular markers have been widely used to identify the genes or QTLs associated with tissue culture. Currently, the development of high-throughput sequencing technology has greatly facilitated the construction of high-density genetic maps. The tissue culture of different plant populations and segregated families, the assessment of tissue culture traits, and the mapping of QTLs have been successfully implemented in different plant species such as *Arabidopsis thaliana* [22,23], *Zea mays* [24,25], *Secale cereal* [26], *Triticum aestivum* [27], *Hordeum vulgare* [28], *Glycine max* [29], *Oryza sativa* [30,31,32], and *Gossypium hirsutum* [33].

Multiple QTLs related to rice callus regeneration ability have been mapped in various populations [20,31,32,34,35,36,37,38,39], but there have been few reports on the mapping or even cloning of specific genes. In 1997, a study reported the mapping of five QTLs for the mean number of regenerated shoots (NRS) per callus and four QTLs for the regeneration rate (RR) using a BC_1_F_5_ (backcrossed with Nipponbare) population from the cross of Nipponbare (*Oryza sativa* ssp. *geng*) and Kasalath (*Oryza sativa* ssp. *xian*) [40]. In 2005, Nishimura et al. identified and cloned a ferredoxin-nitrite reductase (NiR) gene, which can reduce nitrite accumulation and increase RR from a population constructed by the conventional crossing of the hard-to-regenerate rice variety Koshihikari and the easy-to-regenerate rice variety Kasalath [31,35]. In 2009, Zhao et al. used SSR markers in the chromosome segment replacement lines (CSSL) between Nipponbare and Zhenshan 97B to investigate four tissue culture traits (the frequency of callus induction, callus subculture capability, the frequency of plant regeneration, and mean plantlet number per regenerated callus) through two tissue culture protocols, resulting in the identification of 29 QTLs on chromosomes 1, 3, and 10 [32]. In 2013, Li et al. employed SNPs as recombination markers to construct a high-density genetic map of the 93-11 × Nipponbare recombinant inbred line (RIL) and mapped 25 tissue-culture-related QTLs [20]. Accordingly, a genome-wide association study (GWAS) by Zhang et al. on 529 sequenced rice accessions revealed 88 loci correlated to the rate, speed, and time of callus induction, 21 of which were located in previously reported QTLs [39]. Lastly, an allele of *BROWNING OF CALLUS 1* (*BOC1*) from wild rice was found to reduce callus browning in indica cultivars by decreasing cell death and senescence in response to oxidative stress [38].

The difficulty of the genetic transformation of most indica varieties is mostly caused by the poor regeneration ability of their callus [13,41,42,43]. Theoretically, the problem can be solved by finding the optimal conditions for each variety (genotype), which is obviously time-consuming, labor-intensive, and extremely impractical. Genetic analysis based on a common medium may be an effective and practical approach to better locate and utilize the loci related to tissue culture ability. In this study, 300 families of 93-11 × Nipponbare (Nip) high-generation RIL were tissue-cultured, and their callus-regeneration-related phenotypes were investigated for QTL mapping analysis. The objective was to map the genes related to rice callus regeneration ability and improve the culturability of the indica rice varieties.

## 2. Materials and Methods

### 2.1. Genetic Population

The indica rice variety 93-11 is an important restorer line for the utilization of rice heterosis, but its tissue culture and genetic transformation are difficult, which seriously limits its direct application in genetic engineering. The tissue culture and genetic transformation of the japonica rice variety Nipponbare are relatively easy. This study used the 93-11×Nipponbare high-generation (F_20_) RIL constructed by Dr. Han Bin’s team, which included 300 families, and constructed a high-density bin map by resequencing, including 2778 bin markers covering the whole genome.

### 2.2. Tissue Culture

The mature seeds of the RIL (F_20_) and parental 9311 and Nip (302 families in total) were dehulled, and the seeds were sterilized (soaked in 75% alcohol for 30 s, then in HgCl_2_, shaken for 15 min, washed with sterilized single-distilled water repeatedly to remove residual HgCl_2_, and then soaked for 5 h). The induction culture of the callus was carried out using N1 induction medium (based on N6 medium, with the addition of 2.5 mg/L 2,4-D (Sigma, St. Louis, MO, USA), 0.3 g/L proline, 3 g/L phytagel (Sigma, St. Louis, MO, USA), and 30 g/L sucrose; pH 6.0). Each family was inoculated with 40 rice grains in five triangular flasks (8 grains/triangular flask) and cultured in the dark at 28 °C for 35 d until callus growth. Embryogenic callus (callus with compact particles, pale yellow color, and uniform size) was selected and transferred to differentiation medium M1 (based on MS, supplemented with 2.0 mg/L 6-BA (Sigma, St. Louis, MO, USA), 2.0 mg/L KT (Sigma, St. Louis, MO, USA), 0.2 mg/L IAA (Sigma, St. Louis, MO, USA), 0.2 mg/L NAA (Sigma, St. Louis, MO, USA), 1 g/L CH (Sigma, St. Louis, MO, USA), 0.5 g/L L-proline, 3 g/L phytagel, 30 g/L sucrose; pH 6.2) for callus differentiation. Each family was inoculated with five triangular flasks. A total of 20 calli (4/triangular flask) were used as a statistical unit, and they were differentiated and cultured at 26 °C in a light and dark cycle (16 h light/8 h dark). All the above culturing processes were carried out in batches with three biological replicates.

### 2.3. Investigation of Regeneration-Ability-Related Traits

In the regeneration culture, we selected two parameters to measure the regeneration ability: the plant regeneration rate (PRR), which was calculated as the number of calli regenerated to plants/the total number of inoculated calli, and the total green plant rate (TGPR), which was calculated as the total number of regenerated plants/the total number of inoculated calli. From the 4^th^ day of regeneration culture, the total number of calli with green spots in each statistical unit of all families was counted every other day. The time at which each family showed green spots and the first green plantlet was recorded. After 45 d, all regenerated plants in each statistical unit were taken out and counted. Finally, the PRR and TGPR of all materials in three replicates were obtained, producing three sets of six groups of data.

### 2.4. Data Analysis and QTL Positioning

A total of six groups of PRR and TGPR data were obtained to analyze the correlation between repetitions and traits, and Origin (version 9.0, OriginLab Corporation, Northampton, MA, USA) was used to draw a frequency distribution map. The QTL analysis of regeneration-related traits was carried out in Window QTL Cartographer V2.5 (WinQTLCart version 2.5. available online: http://statgen.ncsu.edu/qtlcart/WQTLCart.htm, accessed on 6 December 2022) using composite interval mapping (CIM) [44]. A 10 cM scan window was employed, and the likelihood ratio statistic was computed every 1 cM. A permutation test was conducted 1000 times to determine the experiment-wide significance (*p* < 0.05) thresholds for QTL detection [45]. The location of a QTL was determined according to its LOD peak location and the surrounding region with a 2 LOD confidence interval (≈97%) calculated using WinQTLCart, and the preset LOD value was 3.0.

### 2.5. Construction of NIL

Using the MUS database (Rice Genome Annotation Project. available online: http://rice.uga.edu/, accessed on 6 December 2022) as the reference genome and the published genome sequences of Nipponbare and 93-11 (Epigenomic & Genomic Annotation Database for Rice. available online: http://www.elabcaas.cn/rice/index.html, accessed on 6 December 2022), and by referring to the database information on rice genome variation (Rice Variation Map version 2.0. available online: http://ricevarmap.ncpgr.cn, accessed on 6 December 2022), we screened the sites with the insertion or deletion of more than 20 bp, designed specific primers, and obtained 100–300 bp products through PCR amplification. We performed 4% agarose gel electrophoresis on PCR products to detect polymorphisms between parents and screen for available markers to define the boundaries of QTLs and subsequent near-isogenic lines (NILs).

According to the bin map, the F182 family with better regeneration ability and the crossover occurred near the segment covered by the QTL, the overall genetic background was biased towards 93-11, and BC_3_F_1_ was obtained after three consecutive backcrosses with the parent 93-11. Molecular marker-assisted selection (MAS) was used to select individual plants that were heterozygous in the QTL region and selfed for two generations to obtain BC_3_F_3_.

### 2.6. Effect Verification of QTL

Recombinant plants were selected for this QTL in the BC_3_F_2_-182 population by MAS, and the seeds were harvested to obtain BC_3_F_3_. BC_3_F_3_-182 was first induced with N1 medium. After 35 d, the callus was regenerated with M1 medium. After 45 d, the PRR and TGPR of homozygous-positive and homozygous-negative families were calculated to determine the effect of QTLs. The method was the same as above.

### 2.7. RT-PCR and qRT-PCR

Total RNA was isolated from the rice calli of the regeneration culture using TRIzol Reagent (Invitrogen, Carlsbad, CA, USA) and the phenol-chloroform protocol (acidic phenol: chloroform: isoamyl alcohol = 25: 24: 1). Genome DNA was removed by DNase I (Promega, Madison, WI, USA).

The expression of all genes of *qPRR3* was determined by RT-PCR and qRT-PCR. RT was performed using a transcriptor First Strand cDNA Synthesis Kit (Roche, Mannheim, Germany); RT-PCR was conducted using rTaqTM (Takara, Dalian, China) with a 9700 PCR System (Applied Biosystems, Foster City, CA, USA); RT-PCR products were separated on 1% agarose gel; and qRT-PCR was conducted using SYBR^®^ Premix Ex TaqTM (Takara, Dalian, China) with a 7500 Real-Time PCR System (Applied Biosystems, Foster City, CA, USA).

## 3. Results

### 3.1. Differences in Callus Regeneration Ability

Vitrified and browned non-embryogenic callus has almost no regeneration ability. Here, embryogenic callus with a dry, compact, yellow or light yellowish, and nodular appearance was chosen for study. Generally, when callus is transferred to the regeneration medium, green spots appear within about 7 d, and the regeneration of plantlets occurs after about 20 to 45 d. In this study, the callus of easily regenerated RILs such as F182 and Nip showed rapid proliferation, and green spots appeared after 5–7 d (Figure 1A,G), followed by the formation of numerous shoots and the initiation of tissue differentiation, resulting in the gradual regeneration of plantlets (Figure 1B,C,H,I). Although the callus of some RILs underwent browning and died gradually during the regeneration process, the normal part still displayed green spots, tissue differentiation, and the eventual regeneration of a few plantlets (Figure 1J–L). However, the callus of the RILs that had low regenerability (such as F65 and 93-11) showed no significant proliferation and were browned and died quickly, without the regeneration of any plantlets (Figure 1D–F).

### 3.2. QTL Mapping of Regeneration-Ability-Related Traits

After the deletion of 125 heterozygous RILs without sufficient embryogenic callus in the induction culture or with large differences between repeats during regeneration culture, a total of 175 RILs were retained and subjected to a statistical analysis of PRR and TGPR. The correlation coefficients of three biological replicates were calculated to be 0.925, 0.933, and 0.926 for PRR and 0.86, 0.796, and 0.704 for TGPR, respectively. There were small differences between replicates, and the average was taken for subsequent analysis. Frequency statistical analysis revealed the continuous distribution of these two traits with transgressive segregation. PRR was 63.27 ± 3.33% and TGPR was 355.53 ± 27.24% for Nip. However, 93-11 had extremely low PRR and TGPR (both below 3%) and was particularly difficult to regenerate, with the regeneration of one plantlet from only one replicate. (Figure 2A,B). The correlation coefficient between PRR and TGPR was 0.936, indicating that they were significantly positively correlated (Figure 2C).

With an LOD of 3.0, two QTLs were detected for PRR on chromosome 3 and 5, and one QTL was detected for TGPR on chromosome 3 (Figure 3). Each QTL was named according to rice QTL nomenclature rules [46] (Table 1). *qPRR3* overlapped with *qTGPR3*, both of which were located at Bin653-Bin655 (2.75–3.00 Mb) on chromosome 3. The two QTLs showed a positive additive effect derived from Nip, with LOD values of 7.40 and 9.52, and had the highest PVE values of 13.40% and 17.07%, respectively. In addition, another QTL for PRR was located on chromosome 5, named *qPRR5*, which also showed an additive effect derived from Nip, with an LOD value of 3.89 and a PVE value of 6.73%. It was located at Bin1218-Bin1219 (4.95 Mb–5.25 Mb). *qPRR3* was shared by both PRR and TGPR traits with a PVE value greater than 10%. Hence, *qPRR3* was chosen for follow-up analysis.

### 3.3. Validation of qPRR3 Effect

RIL182 was backcrossed with 93-11 for the construction of an NIL to further verify the effect of *qPRR3* and narrow its interval by increasing the marker density.

RIL182, whose genetic background was 55.04% from 93-11 and 44.96% from Nip, underwent chromosome rearrangement between Bin653 and Bin655 within *qPRR3*, where Bin653 and Bin654 were derived from Nip, and Bin655 was derived from 93-11. The test of regeneration ability revealed that the PRR of RIL182 was 60 ± 5% and the TGPR was 268.33 ± 30.14%. BC_3_F_2_-182 was obtained by the backcrossing of three successive generations with RIL182 as the male parent and 93-11 as the female parent.

Since the seeds of heterozygous BC_3_F_2_ would segregate and cause errors in identification, homozygous families for *qPRR3* were selected for phenotypic examination. By the alignment of the parental SNP differences within *qPRR3*, we found four available InDel markers between Bin653 and Bin655 (Appendix A). Using these four InDel markers to screen the BC_3_F_2_–182 population, a total of four recombination events were resolved, two before Bin653 (BC_3_F_2_-182–31 and BC_3_F_2_-182–46), one between InDel 3–3 and InDel 3–4 (BC_3_F_2_-182–78), and one between InDel 3–4 and InDel 3–5 (BC_3_F_2_-182–69) (Figure 4A). Mature seeds of BC_3_F3 were harvested after field planting (Wuhan, Hubei, China). The callus was obtained by N1 induction medium, and its regeneration-related traits were identified by M1 regeneration medium.

Green spots appeared on the callus of these homozygous-positive families about 8 d after inoculation on the regeneration medium, followed by the initiation of proliferation. At 15 d, many shoots were formed, and finally a large number of regenerated plantlets were obtained at 20 d. The regeneration process of homozygous-negative families was close to that of 93-11. The callus showed no proliferation and quick browning and died after 20 d, and only a few regenerated plantlets were obtained (Figure 4C).

For homozygous-positive families, the PRR values were 55.27 ± 0.88% and 53.17 ± 3.64%, which were not significantly different to those of F182 and Nip, but were significantly higher than that of 93-11. Their TGPR values were 245.11 ± 13.92% and 254.76 ± 14.87%, which were slightly lower than that of Nip. For the homozygous-negative families, the PRR values were 15.93 ± 5.14% and 14.57 ± 3.85%, which were significantly lower than those of the homozygous-positive families and slightly higher than that of parent 93-11. The TGPR values were 21.77 ± 8.46% and 22.93 ± 1.98%, which were very significantly lower than those of the homozygous-positive families, but not significantly different from that of its parent 93-11 (Figure 4B). These results indicated that *qPRR3* is a locus affecting rice regeneration. According to the encrypted InDel markers, it was located in the interval between Bin653 and InDel 3–4 with a physical distance of 163 Kb (2.75–2.91 Mb).

### 3.4. qPRR3 Promotes the Formation of Regenerated Shoots

When the callus of BC_3_F_3_-182–31 was regenerated, it showed faster and more obvious browning than that of BC_3_F_3_-182–78, with the production of fewer regenerated shoots and the regeneration of fewer plantlets. To study the role of *qPRR3* in the regeneration process, the calli of Nip, BC_3_F_3_-182–31, and BC_3_F_3_-182–78 with obvious regenerated shoots after 15 d of regeneration culture were selected to observe the state of shoot regeneration by paraffin section (Figure 5).

Numerous shoot primordia were observed in the callus of Nip, which mostly existed in a clustered state (Figure 5A,B). In the callus of BC_3_F_3_-182–78, a large number of regenerated shoot primordia could be observed, some of which existed in the form of single shoots, and some of which were also clustered (Figure 5C,D). However, in the callus of BC_3_F_3_-182–31, there were only a small number of shoot primordia, all of which existed in the form of single shoots (Figure 5E,F). The total number of single shoots and clustered shoots affected the PRR of the callus, and the proportion of clustered shoots affected the TGPR. The proportion of clustered shoots in the BC_3_F_3_-182–78 callus was slightly lower than that in the Nip callus, which might explain why BC_3_F_3_-182–78 had a lower TGPR than Nip, though the PRR of BC_3_F_3_-182–78 and its parent F182 was not significantly different from that of Nip.

The formation of regenerated shoots was hindered in families without *qPRR3*, while *qPRR3* homozygous-positive families could form a large number of regenerated shoots. Therefore, it can be concluded that *qPRR3* can promote the formation of regenerated shoots.

### 3.5. Candidate Genes of qPRR3

We identified 27 genes in *qPRR3* based on the rice annotation database (RGAP). Nip, 93-11, and BC_3_F_2_-182–78 calli were placed in regeneration culture (three biological replicates); RNA was extracted and reverse-transcribed; and the relative expression levels of 27 genes on *qPRR3* were assessed by qRT-PCR (Appendix A). Two genes were not expressed, five genes had extremely low expression levels, and five genes showed no significant differences between parents in their expression levels. A total of 15 differentially expressed genes were found, nine of which were upregulated in Nip and six of which were downregulated in Nip (Table 2). Therefore, we speculated that the candidate genes should be among the nine genes upregulated in Nip.

## 4. Discussion

### 4.1. From Tissue Culture to QTL Mapping

In this study, the regeneration ability of rice callus was measured based on the PRR and TGPR through the tissue culture of 93-11 × Nip RILs using the Japonica Rice Universal Tissue Culture Protocol. Two new QTLs related to callus regeneration were mapped, one of which was shared by both traits (Table 3). Numerous studies have demonstrated that plant tissue culture is regulated by many factors, including the plant genotype, explant source, explant physiological state, medium composition, and culture process, as well as their interactions [47]. However, tissue culture ability varies greatly among different genotypes of rice. Different phenotypic data are obtained for the same population, which may be affected by the protocols, tissue culture traits, and even the operator of the experiment, resulting in the identification of different QTLs (Table 3).

### 4.2. Tissue Culture Protocol

Theoretically, an optimal tissue culture system can be established for each rice variety. At present, there has been extensive research on the cultivation conditions of japonica rice varieties. Most japonica rice varieties can induce a large number of embryogenic calli on the general induction medium N1 and successfully regenerate on the general regeneration medium M1 to form regenerated plantlets. However, for indica rice, there have been no general tissue culture protocols and genetic transformation systems developed.

In the pre-experiment, we assessed the tissue culture of certain RILs and their parents with the general regeneration medium for japonica (M1) and indica, respectively. With the M1 regeneration medium, most RILs could regenerate normally, while 93-11 was difficult to regenerate even in the optimal medium for indica [11]. Based on the results of the GWAS analysis of the inducibility-related traits of 529 rice germplasm resources in our laboratory [39], and considering the broad adaptability and simplicity of the cultivation protocol, we finally selected a cultivation protocol more suitable for japonica rice to test callus regeneration ability.

### 4.3. Evaluation of Callus Regeneration Ability

Since regeneration is easier for embryogenic callus and extremely difficult for non-embryonic callus, it was better begin our assessment of regeneration ability from the embryogenic callus. After 35 d of induction culture, the vitrified root hairy and loose callus was discarded, and the embryogenic callus with a dry, compact, yellow/light yellowish, and nodular appearance was selected for regeneration culture. Finally, 175 of the 300 families were used for data analysis to ensure the accuracy of the experiment. 

Considering that not all green spots eventually develop into shoots and produce regenerated plants, we chose PRR and TGPR to evaluate the callus regeneration ability, which can directly and intuitively reflect the strength of regeneration. Considering that the PRR of the callus was stabilized after 30 d of regeneration and the TGPR tended to increase from 20 d to 45 d, we chose 45 d as the culture period for regeneration. 

The accurate determination of the phenotype was the most difficult part of this study, since tissue culture response is regulated by many factors. To avoid errors as much as possible, firstly, the regeneration traits were reduced to two key traits. Secondly, a pre-experiment was designed to ensure the feasibility of the experimental scheme. Thirdly, the experimental process was unified, and the tissue culture operation was standardized; in addition, the configuration of the medium and the investigation of the phenotype data were completed at the same time every day. Fourthly, all tissue culture operations and inspections were completed by one author to avoid errors caused by statistical standard variations between operators.

### 4.4. qPRR3 Enhances Rice Regeneration by Promoting Shoot Formation

We chose the PRR and TGPR to study rice callus regenerability, considering that callus regeneration is a complex process. Through QTL mapping, we identified that *qPRR3* on chromosome 3 was shared by both traits, with a positive additive effect from Nip, whose PVE value for the PRR and TGPR was 13.40% and 17.07%, respectively (Figure 3; Table 1). The effect of this QTL was verified in a subsequently constructed NIL (BC_3_F_3_-182), while the QTL was narrowed to an interval of about 160 Kb (Figure 4).

Classical in vitro culture experiments on plant tissues have demonstrated that plant organogenesis is controlled by the balance of exogenous auxins and cytokinins. A high cytokinin/auxin ratio stimulates shoot formation; a low ratio results in the formation of roots; and an intermediate ratio leads to an unorganized state [49]. When callus is transferred to a regeneration medium, cytokinin participates in its regeneration as an exogenous additive, causing an increase in local endogenous auxin, which can enhance the proliferation of local cells, thereby triggering de novo shoot organogenesis [50]. During this process, the cytokinin signal is first received by the cytokinin receptor, followed by the generation of apical meristems. Moreover, cytokinins are also involved in the formation and maintenance of organizing centers, followed by the formation of shoot progenitors; then, the promeristem is developed into a functional shoot meristem with organ primordia, which are further developed into shoots [51,52,53].

All families with homozygous-positive *qPRR3*, such as Nip, F182, and 69 and 78 in BC_3_F_3_-182, showed higher PRR and TGPR values. Both visual observation and paraffin section observation showed that they exhibited the earlier formation of shoot primordia and more regenerated shoots, as well as a higher proportion of clustered shoots and more regenerated plantlets. Families with homozygous-negative *qPRR3*, such as 93-11 and 31 and 46 in BC_3_F_3_-182, showed lower PRR and TGPR values. It was difficult for the callus to form shoot primordia and clustered shoots. Therefore, it can be inferred that *qPRR3* promotes shoot organogenesis, thereby enhancing the regeneration ability of rice callus.

### 4.5. Development of Indica Rice Varieties with Easy Tissue Culture Based on Japonica Rice Universal Medium Using qPRR3 

Generally, the difficulty of indica callus regeneration can be attributed to the accumulation and oxidation of phenolic compounds produced by traumatic stress in the callus, which causes its browning and death [54,55]. The problem of callus browning in regeneration culture may be alleviated by pretreatment, the improvement of the medium and culture conditions, and additional supplements [56,57], but the effect is not satisfactory. Subsequently, researchers have attempted to screen the loci and genes related to callus browning through association analysis and genetic mapping. However, few loci and genes were obtained until 2020, when Zhang et al. discovered the *BOC1* gene, which can alleviate browning and improve the regeneration rate of rice callus [38].

The browning reaction during callus regeneration is a complex and unavoidable process. However, the new QTL *qPRR3* identified in this study can promote the formation of shoots, providing a new direction for reducing callus browning to facilitate indica rice regeneration. When F182 and 93-11 were backcrossed for three generations, families homozygous for *qPRR3* still exhibited a strong regeneration ability. Easy-to-regenerate near-isogenic material in the 93-11 background can be obtained through the further purification of the background by continuous backcrossing. In addition, the problem of indica rice varieties with poor regenerability may be solved by introducing *qPRR3* into the receptors to improve their regeneration ability.

### 4.6. Putative Candidate Genes in qPRR3

Among the 15 genes with significant parental differences in qPRR3, nine were upregulated and six were downregulated in Nip, and similar patterns were observed in near-isogenic lines. These included some inorganic phosphate transporters, histone demethylases, acetyltransferases, heavy-metal-associated domain-containing proteins, and transcription factors Dp (Table 2). Hence, we believe that the candidate genes should belong to the nine genes upregulated by Nip.

## 5. Conclusions

In this work, three QTLs associated with rice callus regeneration ability were identified based on the PRR and TGPR from the 93-11 × Nip recombinant inbred line population. A new QTL, *qPRR3,* was detected for both traits, which could explain 13.40% and 17.07% of the phenotypic variations in PRR and TGPR, respectively. Subsequently, *qPRR3* was narrowed to an interval of approximately 160 kb by cryptographically tagged NILs. The anatomical structure of the regenerated callus of the NILs revealed that *qPRR3* can improve the callus regeneration ability by promoting the regeneration of shoots. This study provides a new direction for improving indica rice regeneration ability by introducing *qPRR3* into the receptors.

## Figures and Tables

**Figure 1 cells-11-04125-f001:**
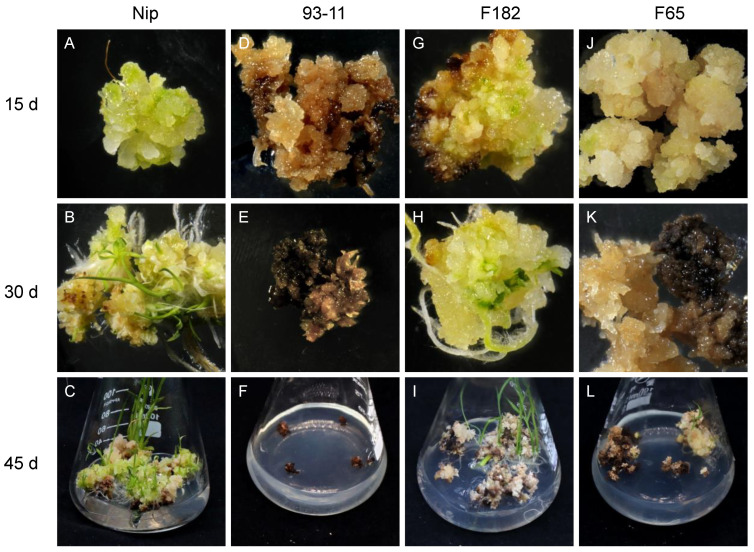
Regeneration of typical RIL. Regeneration of embryogenic callus from parental Nip and 93-11, and RIL of easy-to-regenerate F182 family and difficult-to-regenerate F65 family. (**A**,**D**,**G**,**J**): at 15 d, Nip produced many green spots; 93-11 began to brown; F182 regenerated green spots and began to brown; and F65 only had a few green spots. (**B**,**E**,**H**,**K**): at 30 d, Nip produced regenerated plantlets; 93-11 were browned and died; F182 regenerated shoots; and F65 was half-browned and half-vitrified. (**C**,**F**,**I**,**L**): after 45 d, Nip produced many regenerated plantlets; 93-11 died; F182 produced a few weak regenerated plantlets; and there were no regenerated plantlets for F65.

**Figure 2 cells-11-04125-f002:**
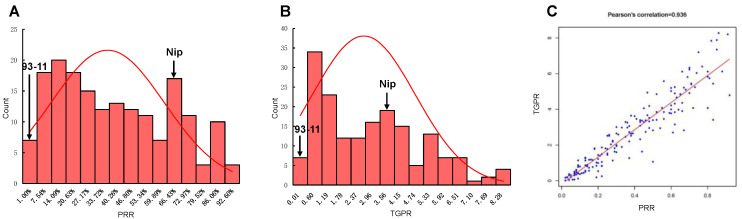
Statistical analysis of PRR and TGPR. (**A**) Frequency distribution diagram of PRR, with arrows indicating the level of the parents 93-11 and Nip. (**B**) Frequency distribution diagram of TGPR in the population, with arrows representing the level of the parents 93-11 and Nip. (**C**) Correlation coefficients between PRR and TGPR.

**Figure 3 cells-11-04125-f003:**
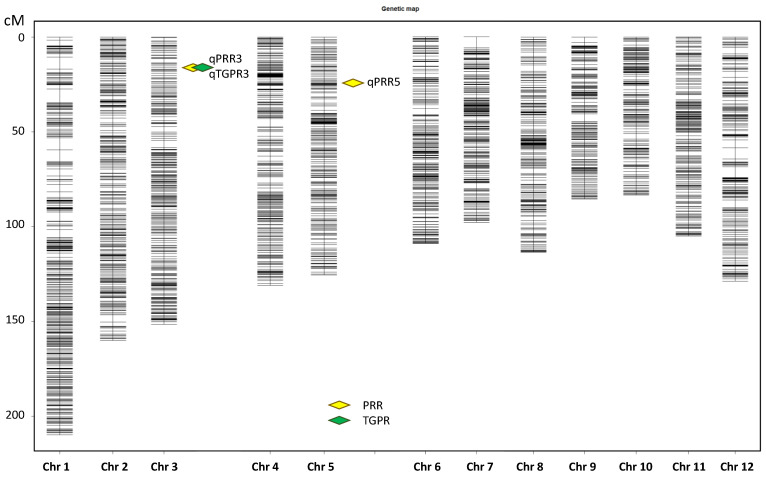
Chromosomal locations of the QTLs.

**Figure 4 cells-11-04125-f004:**
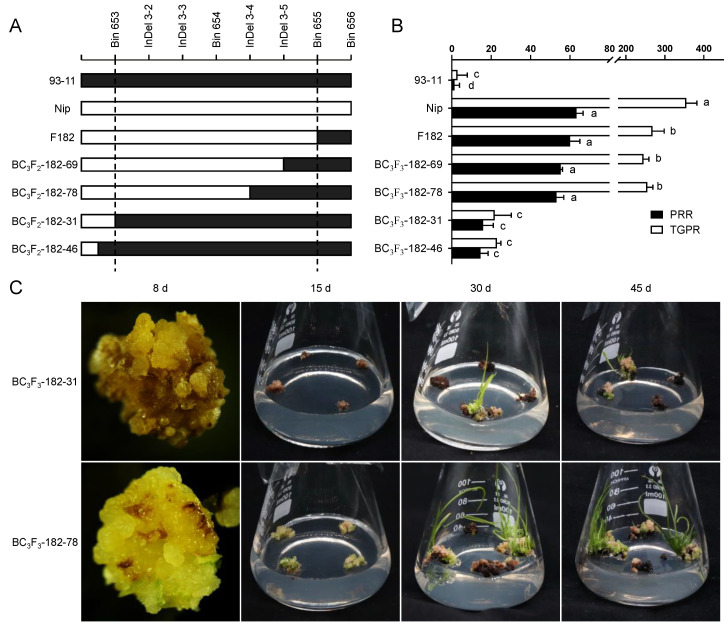
Validation of the effect of *qPRR3*. (**A**) Genotypes of recombinant families in parents, F182, and BC_3_F_2_, with cryptographic markers between Bin653 and Bin655. White and black represent the genotype derived from the Nip and 93-11 allele, respectively, all of which are homozygous families. (**B**) Regeneration phenotypes of all families in A with multiple comparisons. Black represents PRR, and white represents TGPR. BC_3_F_2_-182–69 and BC_3_F_2_-182–78 are homozygous-positive families, and BC_3_F_2_-182–31 and BC_3_F_2_-182–46 are homozygous-negative families. Different letters near the bars denote significant differences among families (*p* < 0.05, ANOVA). (**C**) Upper: regeneration ability of BC_3_F_3_-182 homozygous-negative families. The callus displayed browning and no obvious proliferation at 8 d; only a few plantlets were regenerated and emerged after 30 d; and there was no obvious increase in plantlets until 45 d. Lower: regeneration ability of BC_3_F_3_-182 homozygous-positive families. The callus showed significant proliferation at 8 d; green spots appeared at 15 d; regenerated plantlets were observed at 30 d; and a large number of regenerated plantlets could be observed at 45 d.

**Figure 5 cells-11-04125-f005:**
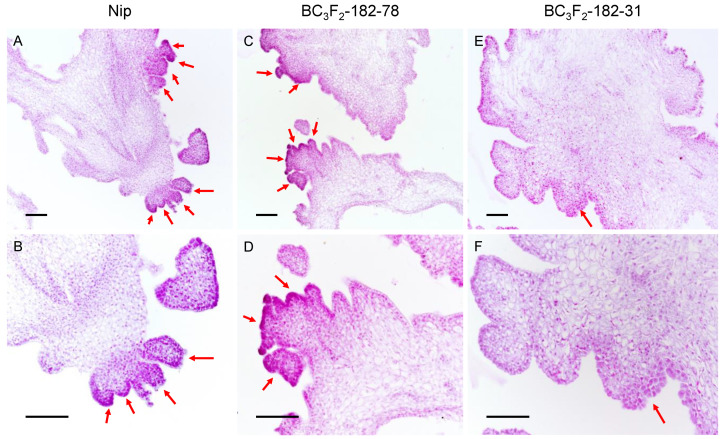
Paraffin section observation of calli after 15 days of regeneration culture. After 15 d of regeneration culture, regenerated shoots appeared. Red arrow: regenerated shoot primordia; top: callus under 10× objective lens; bottom: callus under the same field of 20× objective lens. (**A**,**B**): Nip callus, showing a large number of shoot primordia in the form of clustered shoots. (**C**,**D**): for the callus of the homozygous-positive family BC_3_F_3_-182–78, a large number of shoot primordia coexisted in the form of single shoots and cluster shoots. (**E**,**F**): for the callus of the homozygous-negative family BC_3_F_3_-182–31, there were very few shoot primordia in the form of single shoots. Bar values represent 100 μm.

**Table 1 cells-11-04125-t001:** QTLs identified from the analysis of rice RILs.

QTL	Chr	LOD	Add ^1^	PVE (%)	Bin Marker	Physical Interval (Mb)	Confidence Interval (cM)
*qPRR3*	3	7.395	0.101	13.402	Bin653–655	2.75–3.00	17.08–17.25
*qPRR5*	5	3.890	0.073	6.730	Bin1218–1219	4.95–5.25	25.10–25.43
*qTGPR3*	3	9.520	0.798	17.066	Bin653–655	2.75–3.00	17.08–17.25

^1^ The positive and negative additive effects were from the parent Nip and 93-11, respectively.

**Table 2 cells-11-04125-t002:** Candidate genes of *qPRR3*.

Gene ID	CDS Coordinates (5’–3’)	Expression Level	Gene Product Name
LOC_Os03g05540	2757582–2762244	Nip↓ ^1^	tetratricopeptide repeat-containing protein, putative, expressed
LOC_Os03g05550	2767839–2768531	Nip↑ ^2^	expressed protein
LOC_Os03g05560	2772434–2771187	Extremely low	zinc finger, C3HC4-type domain-containing protein, expressed
LOC_Os03g05570	2777769–2776612	Nip↑	RING-H2 finger protein ATL3F, putative, expressed
LOC_Os03g05580	2793544–2794224	Not expressed	expressed protein
LOC_Os03g05590	2800290–2800924	Extremely low	AP2 domain-containing protein, expressed
LOC_Os03g05600	2802871–2803283	Not expressed	hypothetical protein
LOC_Os03g05610	2804141–2805943	Extremely low	inorganic phosphate transporter, putative, expressed
LOC_Os03g05620	2809678–2807555	Nip↓	inorganic phosphate transporter, putative, expressed
LOC_Os03g05630	2811738–2814268	No significant difference	expressed protein
LOC_Os03g05640	2817367–2815438	Nip↓	inorganic phosphate transporter, putative, expressed
LOC_Os03g05650	2818997–2820125	Extremely low	expressed protein
LOC_Os03g05660	2828096–2822929	Nip↓	appr-1-p processing enzyme family protein, putative, expressed
LOC_Os03g05680	2833133–2837273	Nip↑	histone demethylase JARID1C, putative, expressed
LOC_Os03g05690	2838772–2841691	Nip↑	ZOS3-03—C2H2 zinc finger protein, expressed
LOC_Os03g05700	2842391–2841790	Nip↑	expressed protein
LOC_Os03g05710	2844831–2843447	Nip↑	acetyltransferase, GNAT family, putative, expressed
LOC_Os03g05720	2845709–2851689	No significant difference	WD domain, G-beta repeat-domain-containing protein, expressed
LOC_Os03g05730	2856682–2852293	No significant difference	cell-division control protein 48 homolog E, putative, expressed
LOC_Os03g05740	2860232–2857125	No significant difference	ras-related protein, putative, expressed
LOC_Os03g05750	2868242–2866315	Nip↑	heavy-metal-associated domain-containing protein, putative, expressed
LOC_Os03g05760	2870362–2875430	Nip↑	transcription factor Dp, putative, expressed
LOC_Os03g05770	2878828–2880890	Extremely low	peroxidase precursor, putative, expressed
LOC_Os03g05780	2887922–2883005	Nip↑	4-coumarate—CoA ligase-like 7, putative, expressed
LOC_Os03g05800	2900285–2897476	Nip↓	expressed protein
LOC_Os03g05806	2901596–2906614	No significant difference	pseudouridine synthase family protein, putative, expressed
LOC_Os03g05812	2907108–2912670	Nip↓	expressed protein

^1^ Nip↓, The relative expression of this gene was downregulated in Nip compared with 9311. ^2^ Nip↑, The relative expression of this gene was upregulated in Nip compared with 9311.

**Table 3 cells-11-04125-t003:** QTLs identified from previous studies of regeneration-related ability in rice.

Title 1	Previous Studies					
	QTL Number	Chr. Code Involved	Population Type	Population Size	Mapping Parents ^1^	References
Regenerated shoots	5	1, 2, 4	BC_1_F_5_	98	Nipponbare (J), Kasalath (I)	[40]
	4	3, 7, 12	RIL	150	Nipponbare (J), 93-11 (I)	[20]
Regeneration rate	4	2, 4	BC_1_F_5_	98	Nipponbare (J), Kasalath (I)	[40]
	6	1, 4, 6, 8, 10, 12	CSSL	139	Nipponbare (J), Zhenshan 97B (I)	[32]
	4	2, 3, 11	RI (F_13_:F_14_)	164	Milyang23 (tongil), Gihobyeo (J)	[48]
	4	1, 2, 3, 6	BC_1_F_2_	180	Nipponbare (J), Kasalath (I)	[31]
	2	2, 4	F_2_	79	Norin 1 (J), Tadukan (I)	[37]
	4	2, 3, 7, 12	RIL	150	Nipponbare (J), 93-11 (I)	[20]
	no	no	BC_1_F_3_	183	Koshihikari, Kasalath	[36]
Callus browning	1	3	F_2_	198	Teqing (I), Yuanjiang (Wild)	[38]

^1^ Subpopulations of parents are shown in mapping parents.

## Data Availability

The data that support the findings of this study are available from the corresponding author upon reasonable request.

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
