# Peer review of "QTLs Related to Rice Callus Regeneration Ability: Localization and Effect Verification of *qPRR3"

_cells, 2022, doi:10.3390/cells11244125_

Round 1

Reviewer 1 Report (Previous Reviewer 3)

The study is most well conducted.

Reviewer 2 Report (Previous Reviewer 2)

I have checked that the authors have responded to my comments made in the earlier version of the manuscript in a very good manner. I have no other comments on the MS. The MS is in an acceptable form from my side. 

Reviewer 3 Report (Previous Reviewer 1)

I accept the research article in its present form. Now this article looks like complete story.

This manuscript is a resubmission of an earlier submission. The following is a list of the peer review reports and author responses from that submission.

Round 1

Reviewer 1 Report

1.     Identify genes present in the QTL qPRR3 in Nipponbare and 93-11 and provide list of putative contributors.

2.     Add expression of these genes in both parents and contrasting progeny RILs for PRR.

3.     Include this part with putative functions of the genes in discussion.

Author Response

Reviewer #1

Comments:

  1. Identify genes present in the QTL qPRR3 in Nipponbare and 93-11 and provide list of putative contributors.
  2. Add expression of these genes in both parents and contrasting progeny RILs for PRR.
  3. Include this part with putative functions of the genes in discussion.

Answer: We chose not to take these suggestions for reasons explained below.

We have already identified genes present in the QTL qPRR3 in Nipponbare and 93-11. Parental SNP difference analysis, expression detection and function prediction were carried out on these genes, and several possible candidate genes were obtained. We are investigating the function of this gene, but it’s not finished yet.

Reviewer 2 Report

The MS, "QTLs related to rice callus regeneration ability: localization and effect verification of qPRR3" is a wonderful effort to study one of the least explored traits, probably due to the nature of the trait and the natural variability for this trait viz. regeneration ability.

The novelty includes new mapping population and new traits. However, in spite of that, QTLs detected were too limited and only 15-20% variation could be covered. What other traits in your opinion need to be accounted for to detect more variation through molecular markers. In addition, you also did not went for epistatic QTL detection. I believe that you might detect new QTLs.

There are 2-3 comments made in the attached file which need to be addressed.

Best regards,

Author Response

Reviewer #2

Comments:

1 QTLs detected were too limited and only 15-20% variation could be covered.

2 epistatic QTL detection

Answer: The research in this paper focuses on the discovery and application of key QTLs, hoping to screen candidate genes. We used a higher LOD (LOD=3), which resulted in qPRR3 being highlighted for both traits. But this does not mean that the number of QTLs detected in the study is limited. In fact, if the LOD is reduced to 2.0, we can obtain more variant sites. So, epistasis QTL detection is not necessary.

3 What other traits in your opinion need to be accounted for to detect more variation through molecular markers.

Answer: Callus redifferentiation is a complex process. In addition to the most intuitive plant redifferentiation rate and total green plant rate, traits such as callus proliferation, callus browning, regeneration shoot proliferation speed, and regeneration shoot emergence time can be used for QTL detection. Based on the different tissue culture protocols than previous ones, we can definitely identify some new loci.

4 There are 2-3 comments made in the attached file which need to be addressed

Answer: Line 58: Done. Please see Line 58. Line 70: Done. Please see Line 70. Line 80: Done. Please see Line 80. Line 105: Done. Please see Line 105. Line 109: Done. Please see Line 109. Line 123: Done. Please see Line 123. Line 193: Done. NIL was first explained on line 140.

Line 385: delete “completely”. The callus regeneration rate is a quantitative trait, which is controlled by multiple genes. It is impossible to completely solve it by introducing a gene locus. With the complete explanation of the genes controlling the trait, this problem can be finally solved. However, this research cloned a gene locus with the largest effect, which can improve the problem of callus regeneration to a certain extent.

Reviewer 3 Report

The manuscript entitled " QTLs related to rice callus regeneration ability: localization and effect verification of qPRR3 " reported the story about QTL mapping of plant regeneration rate (PRR) and total green plant rate (TGPR) using RILs derived from the crossing 93-11 and Nip. This MS is of interest, and is important to identify new genes for PRR and TGPR. The study is well conducted, However, there are a few points that I hope useful for the authors to further improve the paper as below:

1 The sentence in Line 144 “Molecular marker-assisted (MSA) selection” may be “MAS”

2 The sentence in Line 214-217 should be more accurate presentation like the previously paper (Fan et al., 2006 and Bai et al., 2017).

3 The mapping results in Fig4 is contradiction with the presentation in Line 255-257

4 The order of Fig5 should adjust according to the order of appearance in MS.

Author Response

Reviewer #3

Comments:

1 The sentence in Line 144 “Molecular marker-assisted (MSA) selection” may be “MAS”

Answer: Done. Please see Line 144 and Line 419

2 The sentence in Line 214-217 should be more accurate presentation like the previously paper (Fan et al., 2006 and Bai et al., 2017).

Answer: Done. Please see Line 214 to 217. “Using these four InDel markers to screen the BC3F2-182 population, a total of 4 recombination events were resolved, two before Bin653 (BC3F2-182-31 and BC3F2-182-46), one between InDel 3-3 and InDel 3-4 (BC3F2-182-78) and one between InDel 3-4 and InDel 3-5 (BC3F2-182-69) (Figure 4A).”

3 The mapping results in Fig4 is contradiction with the presentation in Line 255-257

Answer: Done. Please see Line 256

4 The order of Fig5 should adjust according to the order of appearance in MS.

Answer: Done. Please see Fig 5

Round 2

Reviewer 1 Report

Authors are not ready to consider any comments given.